# Freehand 1.5T MR-Guided Vacuum-Assisted Breast Biopsy (MR-VABB): Contribution of Radiomics to the Differentiation of Benign and Malignant Lesions

**DOI:** 10.3390/diagnostics13061007

**Published:** 2023-03-07

**Authors:** Alberto Stefano Tagliafico, Massimo Calabrese, Nicole Brunetti, Alessandro Garlaschi, Simona Tosto, Giuseppe Rescinito, Gabriele Zoppoli, Michele Piana, Cristina Campi

**Affiliations:** 1Dipartimento di Radiodiagnostica, IRCCS—Ospedale Policlinico San Martino, Largo Rosanna Benzi 10, 16132 Genvoa, Italy; 2Dipartimento di Scienze della Salute (DISSAL), Università di Genova, Via L.B. Alberti 2, 16132 Genova, Italy; 3Dipartimento di Medicina Sperimentale (DIMES), Università di Genova, Via L.B. Alberti 2, 16132 Genova, Italy; 4Dipartimento di Medicina Interna (DIMI), Università di Genova, v.le Benedetto XV 6, 16132 Genova, Italy; 5Dipartimento di Matematica (DIMA), Università di Genova, Via Dodecaneso 35, 16146 Genova, Italy; 6Life Science Computational Laboratory (LISCOMP), IRCCS Ospedale Policlinico San Martino, 16132 Genova, Italy

**Keywords:** breast cancer, MR-guided vacuum-assisted breast biopsy, radiomics, magnetic resonance imaging

## Abstract

Radiomics and artificial intelligence have been increasingly applied in breast MRI. However, the advantages of using radiomics to evaluate lesions amenable to MR-guided vacuum-assisted breast biopsy (MR-VABB) are unclear. This study includes patients scheduled for MR-VABB, corresponding to subjects with MRI-only visible lesions, i.e., with a negative second-look ultrasound. The first acquisition of the multiphase dynamic contrast-enhanced MRI (DCE-MRI) sequence was selected for image segmentation and radiomics analysis. A total of 80 patients with a mean age of 55.8 years ± 11.8 (SD) were included. The dataset was then split into a training set (50 patients) and a validation set (30 patients). Twenty out of the 30 patients with a positive histology for cancer were in the training set, while the remaining 10 patients with a positive histology were included in the test set. Logistic regression on the training set provided seven features with significant *p* values (<0.05): (1) ‘AverageIntensity’, (2) ‘Autocorrelation’, (3) ‘Contrast’, (4) ‘Compactness’, (5) ‘StandardDeviation’, (6) ‘MeanAbsoluteDeviation’ and (7) ‘InterquartileRange’. AUC values of 0.86 (95% C.I. 0.73–0.94) for the training set and 0.73 (95% C.I. 0.54–0.87) for the test set were obtained for the radiomics model. Radiological evaluation of the same lesions scheduled for MR-VABB had AUC values of 0.42 (95% C.I. 0.28–0.57) for the training set and 0.4 (0.23–0.59) for the test set. In this study, a radiomics logistic regression model applied to DCE-MRI images increased the diagnostic accuracy of standard radiological evaluation of MRI suspicious findings in women scheduled for MR-VABB. Confirming this performance in large multicentric trials would imply that using radiomics in the assessment of patients scheduled for MR-VABB has the potential to reduce the number of biopsies, in suspicious breast lesions where MR-VABB is required, with clear advantages for patients and healthcare resources.

## 1. Introduction

Breast Magnetic Resonance Imaging (MRI) is widely recognized as one of the best methods for diagnosing breast diseases, not only after conventional breast imaging such as mammography, digital breast tomosynthesis or ultrasound, but also as a stand-alone modality for breast cancer screening in high-risk women [1,2,3]. In addition, breast MRI can be used for image-guided needle biopsy for MRI-only visible lesions. Breast MRI can show several lesions not clearly visible with standard imaging but which are Breast-Imaging Reporting and Data System BI-RADS 4–5 lesions where there is an indication to perform a biopsy [1,2,3]. Ultrasound is the preferred modality for identification/biopsy of lesions detected at first with breast MRI. In these cases, the so-called second-look targeted ultrasound can detect lesions and then serve as a guide to biopsy for up to 60% of lesions detected at first with breast MRI [4]. If the lesion is visible with ultrasound, even at second-look ultrasound, the biopsy is performed under ultrasound guidance because the procedure is faster, more widely available and relatively cheaper compared to MR-guided biopsy [1,2,3]. The European Society of Breast Imaging (EUSOBI) estimated that MRI-guided biopsy is performed by only a minority of European breast radiologists (35.4%), either using MRI for wire-localization or needle biopsies [1], and more often in the academic environment [1]. Therefore, MRI-guided biopsy must be considered as a specialist procedure with dedicated training and sampling equipment and a long execution time compared to diagnostic MRI or ultrasound-guided biopsy. Moreover, it has been estimated that around 50–70% of MRI findings requiring biopsy turn out to be non-cancer cases [5]. In recent years, radiomics and artificial intelligence have been increasingly applied in breast MRI [6], not only to classify breast lesions or predict response to treatment but also to differentiate benign from malignant lesions [6,7,8,9,10,11,12,13,14,15,16,17,18,19,20,21,22,23,24,25,26]. Indeed, radiological practice includes a qualitative and subjective evaluation of the visible lesion describing different characteristics of the tumor aspect (e.g., the presence of spiculations and the presence of necrosis and microcalcifications). In addition, using MRI it is possible to assess the type of enhancement and the anatomic relationships to the nipple-areola complex and the pectoralis muscle and other surrounding tissues to guide further treatment. In recent years, it has become evident that new advances in radiomics and artificial intelligence result not only in early diagnosis of breast cancer but also in characterization of the lesion on radiological images with the goal of obtaining tailored diagnoses and treatments (an approach described as ‘personalized medicine’ as it aims to offer the best treatment available to the right patient at the most appropriate time) [26]. In this emerging clinical setting, quantitative use of standard medical MR images is the obvious evolution towards personalized medicine. Advancements in medical image analysis through the application of artificial intelligence methods can process large quantities of images, obtaining data beyond the normal capabilities of the human eye. The perspective of this rather modern approach to image interpretation relies on the assumption that mutations have an effect on image features and that image features can be used to train and feed neural networks for predicting disease follow-ups. However, the advantages of using radiomics to evaluate lesions amenable to MR-guided vacuum-assisted breast biopsy (MR-VABB) are unclear. Given the known difficulties in performing breast biopsies under MRI guidance, the goal of the present study is to investigate the potential of radiomics using dynamic contrast-enhanced MRI (DCE-MRI) for identification of malignant and benign lesions among those scheduled for MR-VABB, in order to potentially improve standard radiological assessment and reduce the number of biopsies.

## 2. Materials and Methods

### 2.1. Patients

This study is a subsidiary of a prospective monocentric study that included both symptomatic and asymptomatic women, including women with clinical findings or high-risk patients with BI-RADS breast density C and D [NCT03033030]. Patients who underwent standard radiological assessment were included when MRI-only visible lesions were scheduled for MR-VABB. All MRI-only visible lesions were scheduled for MR-VABB if second-look ultrasound was not able to find a lesion to be biopsied. All patients agreed to undergo MR-VABB when indicated. Contrast-enhancement mammography was not available in the Radiological Department when data collection was conducted. The original study recruited patients prospectively, but for the radiomics analysis of MR-VABB lesions the data were collected retrospectively. Apart from the study protocol, the most frequent indications for breast MRI were, based on European guidelines and recommendations [2,3,27], patients with a newly diagnosed invasive cancer or patients at high risk for breast cancer or under 60 years of age with discrepancy in size > 1 cm between mammography or tomosynthesis and ultrasound. We kept, as per the institutional protocol, total treatment delay due to pre-operative MRI and MR-VABB workup shorter than 1 month. Therapeutic planning was decided for every patient by a multidisciplinary team called the disease management team (DMT) and composed of oncologists, pathologists, radiation oncologists, radiologists and surgeons.

### 2.2. MR Imaging Technique

All MRI examinations and subsequent biopsies were performed using a clinical 1.5T MRI scanner (Siemens Magnetom Aera 1.5 Tesla, Siemens Healthcare, Enlargen, Germany). The MRI standard protocol included the following sequences: localizing sequence; fat-saturated T2-weighted; diffusion-weighted images performed before contrast-agent injection using a single-shot echo-planar imaging (SE-EPI) sequence (TR/TE = 2496/71 ms, slice thickness = 5 mm, slice spacing = 1 mm, b value = 0/800 s/mm^2^); three-dimensional fat-suppressed gradient-echo T1-weighted sequences (T1-flash3D, 4.33 ms repetition time/173.52 ms echo time; 768 × 768 matrix); 340 × 340 mm field of view with dynamic field of view; 0.9 mm isotropic resolution; acquisition time < 10 min; echo train length: 1; number of excitation: 1 before and after bolus injection of gadolinium-based contrast medium (gadobenate dimeglumine, MultiHance: Bracco Imaging, Milano, Italia; 0.1 mmol/L per kilogram of body weight; injection rate of 2.0 mL/s followed by 20 mL saline flush). On axial planes, subtraction images were obtained from the contrast-enhanced and unenhanced images; then, maximum-intensity-projection images were reconstructed using the first subtracted contrast-enhanced dynamic sequence.

The American College of Radiology BI-RADS lexicon [28] was used as a reference for reporting and description of all breast MRI findings and for selection of lesions to be biopsied when second-look ultrasound was not able to demonstrate any lesion. Second-look ultrasound was always performed by radiologists with sub-specialization in breast imaging and at least 10 years’ experience (M.C, A.G., G.R., S.T.). Contrast-enhanced mammography was not considered an option due to the fact that these lesions were MR-only visible after standard imaging.

### 2.3. Biopsy Technique

The Mammotome^®^ VABB system (Ethicon Endo-Surgery), equipped with a dedicated breast biopsy coil, was used. Around 12–24 samples were taken with a 9G needle. The Mammotome procedure is a minimally invasive procedure known to be safe, effective and well-tolerated for breast tissue sampling. The Mammotome Breast Biopsy System was introduced in 1996 and approved by the US Food and Drugs Administration in 2004. To date, it is considered the standard approach for breast tissue sampling. The Mammotome biopsy system allows radiologists to perform biopsies under US, stereotactic or even MRI guidance with the aim of sampling a sufficient volume of tissue for a highly accurate and specific pathological diagnosis with few complications, even with a large sample. In our study, the number of samples depended on lesion size, breast size and location of the lesion. After sampling, the breast tissue was fixed in buffered formalin; then, an MRI-compatible clip was placed in the biopsy site. Informed consent for the VABB procedure was obtained from all the patients included in the present study. Patients who refused to provide informed consent, who were intolerant of or allergic to the local anesthetic (lidocaine of mepivacaine), or who had clinical signs of active skin infections on the breast to be sampled were considered ineligible for a biopsy. Bleeding during the intervention and post-interventional hematoma were treated promptly, if present, by dedicated staff. Pressure dressing was applied to prevent excessive bleeding from the biopsy during the first 24 h after the intervention. The procedure was performed by three experienced radiologists with strong track records in breast imaging and intervention (A.G. and S.T. with 12 years of experience in breast biopsy, G.R. with 14 years of experience in interventional radiology and breast biopsy). On average, the procedure had a duration of 20 min and the mean number of tissue cores removed was 18 (range, 12–24).

### 2.4. Image Segmentation

MR images to be analyzed with radiomics were selected before MR-VABB. The first acquisition of the multiphase DCE-MRI sequence was selected for image segmentation when it had the highest peak of the enhanced phase in accordance with the time-intensity curve. Otherwise, we selected the MRI acquisition with the highest peak of the enhanced phase (the first one of the second sequence when the highest peak was not reached at first). These MRI images allowed the use of high-spatial-resolution imaging permitting precise anatomical definition and morphological delineation of the pathological tissue due to the use of the gadolinium-based contrast medium that highlights hypervascular breast tissue. An open-source platform (3D Slicer software, Version 4.8) [29] was used to segment the pathologically enhancing breast lesion, the process being carried out by one radiologist with 12 years of breast-imaging diagnosis experience (A.T.) under the supervision of an expert with 15 years of experience in image analysis (C.C.). Both readers were blinded to clinical information and to histopathological information on the breast tumor. The volume of interest covering the suspicious tumor (tumor Volume of Interest, VOI) that underwent biopsy was manually delineated slice by slice, and edge-based segmentation offered by the software was used. Although the Region of Interest (ROI) drawn by experts is considered the gold standard for tumor segmentation, we performed a double check by drawing a ‘bounding box’ (a polygon manually drawn that is larger than the tumor and does not include specific contours) and using a thresholding method based on MRI contrast enhancement on the first postcontrast subtracted image. This procedure allowed precise and double-checked determination of the boundaries of the tumors. When threshold values were chosen for voxels with signal intensity above that threshold in the postcontrast image, those voxels were considered part of the lesion amenable to radiomics analysis. This thresholding technique was performed for every patient, with manual adjustment when necessary. An automated thresholding approach was not implemented in this study. One final point is that the highest peak of the enhanced phase in accordance with the time-intensity curve was selected from subtracted images to differentiate the lesion with high contrast, reflecting the presence of neovascularization typical of breast lesions. The regions of interest were placed into the area of the lesion where the enhancement was strongest in the first non-subtracted postcontrast image. When this was not possible, we selected the MRI acquisition with the highest peak of the enhanced phase (the first one of the second sequence when the highest peak was not reached at first). This method is consistent with the recent literature [30,31].

The workflow of segmentation is shown in Figure 1.

### 2.5. Radiomics Feature Extraction, Selection and Model Development

Radiomics features were extracted using Pyradiomics (3.0.1), which is an open-source Python package interface provided as the ‘Radiomics’ extension for 3D Slicer. The parameters were as follows: ‘resampledPixelSpacing’: [1, 1, 1]; ‘bandwidths’: 25. A total of 93 radiomics features were extracted from raw VOI images with only adjustments of contrast and no imaging filters. The radiomic features were divided into different groups, including shape, first-order statistics and texture features: comprehensive definitions, accurate descriptions and subdivisions into classes of radiomics features are available in the literature [7,8,9,10,11,12,13,14,15,32]. In summary, two- and three-dimensional shape features include descriptors of the size and geometric shape of the ROI; first-order statistics describe the distribution of the voxel intensities throughout the ROI; and texture features refer to the properties of the gray level values. The statistical analysis was performed using R software [32], applying the Pearson correlation coefficient method to reduce and exclude redundant features. Figure 2 shows the correlation matrix of the original features. Highly correlated features (correlation values smaller than −0.6 and larger than 0.6) were excluded. The dataset was then split into a training set (50 patients) and a test set (30 patients). In order to maintain some uniform balance between classes in the two sets, 20 out of the 30 patients with positive histology for cancer were in the training set, while the remaining 10 patients with positive histology were inserted in the test set, for an overall balance ratio of 0.4 for the training set and 0.34 for the test set.

Then, we performed a logistic regression on the training set in order to identify the features more prominent in the classification of histologically positive and negative subjects. Penalized logistic regression has been designed as an ‘ad hoc’ method for performing feature selection while realizing a classification task for dichotomous, categorical data. This is a maximum-likelihood method that allows the estimation of model parameters while best fitting the data. In general, logistic regression admits two possible penalty terms, i.e., either l2- or l1-norm penalization of the variables. In this study we applied l1-norm penalization, which automatically identifies the features that mostly impact the stratification process. Therefore, in this implementation, logistic regression also computes feature ranking when applied to the test set.

Finally, the area under the curve (AUC) was computed to assess the performance of the model relative to the radiological readers and radiomics, with final surgery as a reference standard. After 6 months of wash-out, three breast radiologists (G.R. with 15 years of experience in breast MRI, S.T. with 12 years of experience in breast MRI and A.G. with 13 years of experience in breast MRI) evaluated in consensus the lesions that were biopsied, blindly to the biopsy results, to assess whether they were malignant or benign, and therefore to compare the radiological results with the radiomics ones.

## 3. Results

A total of 80 patients (80 women) with a mean age of 55.8 years ± 11.8 (standard deviation, SD) were included. The characteristics of the patients in the training set and the validation dataset are reported in Table 1. In our center, B3 lesions underwent surgical excision but, for the purposes of the analysis, were not included among the malignant lesions.

Logistic regression automatically performed redundancy reduction by selecting seven features that were considered for further analysis. Specifically, the logistic regression analysis applied to the training set provided seven features with significant *p* values (<0.05). Table 2 reports the diagnostic performance of the logistic regression model, whereas Table 3 reports the diagnostic performance of the radiological evaluation. The features considered for further analysis were the following: (1) ‘AverageIntensity’, (2) Autocorrelation’, (3) ‘Contrast’, (4) ‘Compactness’, (5) ‘Standard Deviation’, (6) ‘MeanAbsoluteDeviation’, (7) ‘InterquartileRange’.

‘AverageIntensity’ measures the average gray level intensity within the ROI.‘Autocorrelation’ is a texture feature that computes the magnitude of the fineness and coarseness inside the ROI.‘Contrast’ measures the local intensity variation, thus promoting values away from the diagonal (i.e., a larger value correlates with a greater disparity in intensity values among neighboring voxels).‘Compactness’ measures to what extent the shape of the tumor is compact relative to a sphere, which is considered to be the most compact shape.‘Standard Deviation’ measures the dispersion from the average intensity.‘MeanAbsoluteDeviation’ measures the mean distance of all intensity values from the average of the image pixels.‘InterquartileRange’ measures the difference between the 75th and 25th percentiles of the image array.

These seven features were then used to create a logistic regression model, providing the receiving operating characteristic (ROC) curve of the training set that is shown in Figure 3. The model was then applied to the test set. Table 2 contains the results for both the training and test sets. Table 3 reports data for the radiological readers.

## 4. Discussion

This study shows that radiomics applied to DCE-MRI images was able to increase the diagnostic accuracy of standard radiological evaluation for MRI suspicious findings in women scheduled for MR-VABB. Specifically, an AUC of 0.86 (95% C.I. 0.73–0.94) for the training set and 0.73 (95% C.I. 0.54–0.87) for the test set was obtained by our radiomics regression model. Two other radiomics-based studies [33,34] obtained AUC values in the ranges 0.48–0.59 and 0.80–0.90, respectively. However, in the analysis performed by those studies there was no separation between training and test sets, which typically leads to higher AUC performances. Radiological evaluation of the same lesions scheduled for MR-VABB, on the contrary, was not able to reach acceptable AUC values: 0.42 (95% C.I. 0.28–0.57) for the training set and 0.4 (0.23–0.59) for the test set. The main difference between the radiological evaluation and the radiomics model is due to the notably increased specificity achieved by the latter one. Indeed, radiomics specificity was more than one order of magnitude bigger for the training set and more than four times bigger for the test set with respect to radiological specificity. Sensitivity values were good for both the radiomics model and the standard radiological evaluation, with slightly higher values for the latter. However, one should take into account the fact that all the evaluated lesions were referred for breast biopsy under MRI guidance. Therefore, this outcome concerning sensitivity improves the consistency of our dataset by confirming that the lesions that underwent MR-VABB were suspicious for malignancy according to standard radiological assessment. As a consequence of the results for specificity and sensitivity, the Youden’s Index values provided by radiomics were significantly higher than those provided by radiologists in the case of both the training and the test sets.

The performance of the radiomics model as regards positive predictive value (PPV) and negative predictive value (NPV) was better than in the case of radiological assessment. PPV and NPV are related to disease prevalence (cancer/non-cancer ratio 8/3, 37.5% cancers), which implies that these data always need to be evaluated in this context. NPV is inversely correlated with prevalence, and this study’s cohort already presents a relatively high disease prevalence of 37.5%, consistently with other studies in the literature [35,36,37].

The application of radiomics and artificial intelligence in breast cancer evaluation is rapidly progressing [27,38]. However, the literature lacks studies of radiomics applications in patients scheduled for MR-VABB. The results of this study could be considered to support the application of radiomics in clinical practice to potentially reduce the number of MR-VABB analyses. Indeed MR-VABB is considered as a specialist procedure that needs dedicated training, even among breast imaging specialists; further, it requires dedicated sampling equipment and a long execution time compared to standard diagnostic MRI or even to ultrasound-guided biopsy. In addition, around 50–70% of MRI findings requiring biopsy turn out to be non-cancer cases; therefore, the potential application of radiomics to lesions scheduled for biopsy could reduce, at least in theory, the number of false positives and eventually the number of biopsies.

We point out that at least three features among those that are useful for radiomics analysis, i.e., ‘AverageIntensity’, ‘Autocorrelation’ and ‘Contrast’, could be related to the contrast media uptake due to the angiogenesis typical of breast cancer. Indeed, ‘AverageIntensity’ is a first-order feature describing the distribution of voxel intensities within the image that could vary according to the increased contrast media uptake. ‘Autocorrelation’ is a measure of the magnitude of the fineness and coarseness of texture, which is likely to be affected by the different distribution of contrast media inside the target lesion. ‘Contrast’ measures the local intensity variation inside the lesion where larger values correlate with a greater discrepancy in intensity values among neighboring voxels, possibly reflecting the contribution of gadolinium inside the lesion. It is more difficult to qualitatively and quantitatively assess the biological correlation between the other features (i.e., ‘compactness’, ‘StandardDeviation’, ‘MeanAbsoluteDeviation’ and ‘InterquartileRange’) and the MRI signal.

However, the biology of breast cancer and its detection on breast MRI is strictly linked to the presence to neovascularization, and it is likely that radiomics features vary in the region of interest due to the injection of gadolinium and can detect changes in the breast tissue beyond the capabilities of the human eye, even for an experienced breast radiologist. For this reason, the results of this study suggest that the radiomics model could perform better than the qualitative evaluation made by radiologists. In addition, the radiomic logistic regression model could noninvasively predict the likelihood of malignancy of breast lesions amenable to breast biopsy. Although in the medical literature many studies are devoted to assessing the possibility of differentiating between benign and malignant lesions for MRI data [7,8,9,10,11,12,13,38,39,40,41,42,43,44,45], there are currently no public and widely accepted radiomics-based guidelines for the pre-operative prediction of malignancy likelihood in patients amenable to MR-VABB. Some recent studies have paved the way to a radiomics-driven exploratory research phase [33,34], and much effort should be made to realize translation into clinical settings. In general, the results of this study should be compared to data obtained in other centers and in larger multicentric datasets, and the study used as a proof-of-concept.

This study has some limitations. First, the sample size is probably not optimal for this kind of study, since most published studies include larger datasets, especially when a large number of features is considered. However, this study has strict clinical inclusion criteria; indeed, we focused only on patients undergoing MR-VABB, who make up a very small portion of all patients undergoing breast MRI in our center. From this perspective, for this population our dataset can be considered large enough (the minimum caseload for a breast radiologist suggested by the European Society of Breast Cancer Specialists (EUSOMA) is 50 guided interventions per year as a whole, and MR-VABB cases are only a small part of them [45,46,47]). We also point out that increasing the sample size could imply the need for multicentric studies, and this would pose data homogenization issues. A multicentric study would add variability due to the use of different MRI scanners and, even, slightly different clinical inclusion criteria for patients submitted to MR-VABB. In addition, slight differences in imaging protocols regarding imaging acquisition parameters could lower the advantages of multicentric studies. However, in the near future, the data and results from this study should be confirmed in large trials to increase and hasten progress on the path toward clinical availability.

Second, image segmentation includes automatic, semi-automatic and manual methods, but image-segmentation algorithms and manual methods still need to be fully evaluated to reduce variability, and, more generally, the quantitative assessment of the reliability of radiomics features with respect to segmentation accuracy is still an open issue to be addressed [48]. In this study, we decided to perform a relatively time-consuming procedure to assure the most accurate segmentation, with the assistance and supervision of radiologists with extensive experience in image analysis and breast imaging. We acknowledge that this approach is time-consuming and could be speeded up by using automatic segmentation and thresholding methods. However, these methods have still to be fully validated, especially in patients undergoing MR-VABB, where lesions, including non-mass like lesions, could have a different appearance on MRI. Third, we do not know the impact on radiomics data of lesion intravisit registration (e.g., aligning data collected within one scan) and lesion intervisit registration (aligning datasets across different scanning sessions).

In conclusion, in this study a radiomics logistic regression model applied to DCE-MRI images increased the diagnostic accuracy of standard radiological evaluation of MRI suspicious findings in women scheduled for MR-VABB. Confirming this performance in a study involving large multicentric trials would imply that radiomics used in the assessment of patients scheduled for MR-VABB has the potential to reduce the number of biopsies, in suspicious breast lesions where MR-VABB is required, with clear advantages for patients and healthcare resources.

## 5. Conclusions

This study investigates, probably for the first time, the effectiveness of a radiomics approach for the identification of malignant and benign lesions among those scheduled for MR-VABB. These preliminary results seem to show that radiomics could outperform standard radiological assessment, thus reducing the number of biopsies. The translation of this approach to the clinical workflow would require, first of all, the validation of this preliminary outcome, for instance via multicentric studies, via a systematic quantitative assessment of the radiomics features’ reliability and via comparison of performances among the possible regression models. Further research would be useful to fill this gap in the literature and to influence clinical practice, especially due to the fact that MR-VABB is a difficult procedure performed by only a minority of radiologists and requiring dedicated training, accurate tissue sampling and expensive equipment. In addition, the procedure is time-consuming compared to ultrasound-guided biopsy. Therefore, if successful, radiomics could have the potential to reduce the number of biopsies, in cases where MR-VABB is required, with clear advantages for patients and healthcare resources.

## Figures and Tables

**Figure 1 diagnostics-13-01007-f001:**
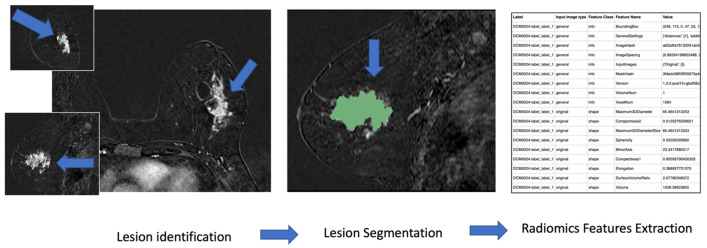
Workflow of image segmentation.

**Figure 2 diagnostics-13-01007-f002:**
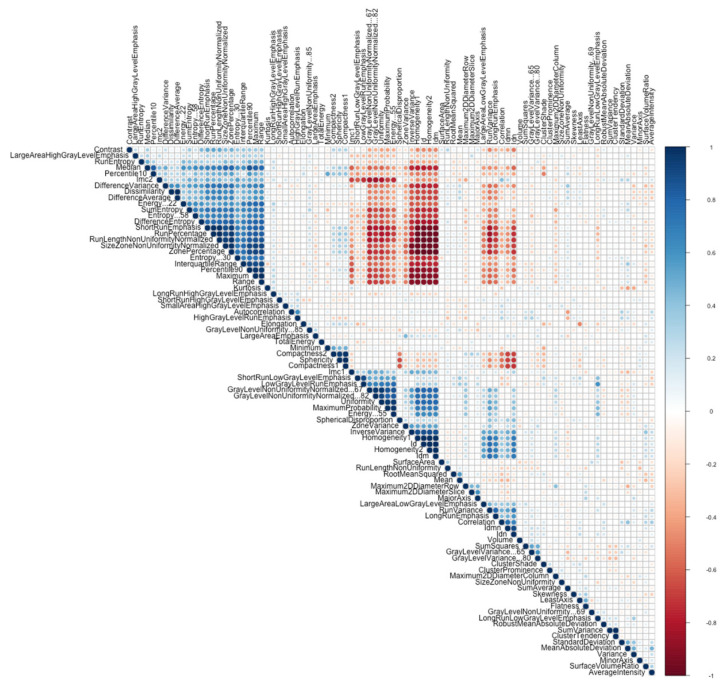
Heatmap showing the correlation values between pairs of radiomic features.

**Figure 3 diagnostics-13-01007-f003:**
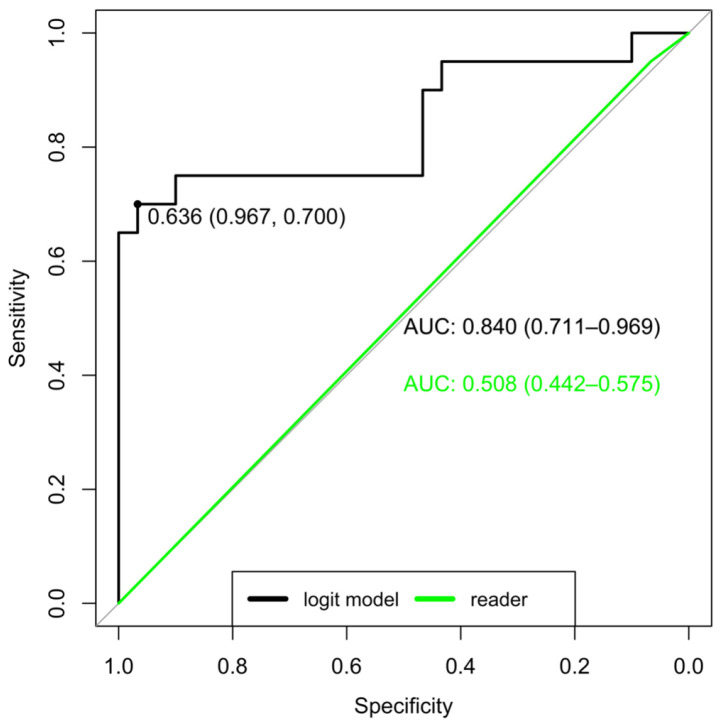
ROC curve (black) for the logistic regression model and ROC curve (green) for the radiological readers.

**Table 1 diagnostics-13-01007-t001:** Clinical and Pathologic Characteristics of Patients for Radiomics Model Construction. * Data are mean ± SDs, with ranges in parenthesis. ** For patients with positive histology. ^Ψ^ Kruskal–Wallis H test was performed. ns: not significant. DCIS: ductal carcinoma in situ. ADH: atypical ductal hyperplasia. LIN: lobular intraepithelial neoplasia. FEA: flat epithelia atypia.

Characteristic	Training Dataset (*n* = 50, 20 Cancers)	Validation Datasets (*n* = 30, 10 Cancers)	*p* Value
Age (Y *)	56 (11.6)	56 (11)	ns
Tumor size ** (cm) ^Ψ^	17 (14)	18 (15)	ns
RM BI-RADS 3	3	1	ns
RM BI-RADS 4	13	6	ns
RM BI-RADS 5	4	3	ns
Cancer subtype			
Ductal invasive	14	7	
Lobular invasive	4	1	
DCIS	2		
Linfoma		1	
Cribriform		1	
ADH	3	2	
LIN	1	2	
FEA	1	1	
Other benign	6	8	

**Table 2 diagnostics-13-01007-t002:** Skill scores for the logistic regression model.

	Training Set (95% CI)	Test Set (95% CI)
Accuracy	0.86 (0.73, 0.94)	0.73 (0.54, 0.87)
Sensitivity	0.70 (0.46, 0.88)	0.40 (0.12, 0.74)
Specificity	0.97(0.83, 1.00)	0.90 (0.68, 0.99)
Youden’s Index	0.67 (0.29, 0.88)	0.30 (−0.20, 0.73)
PPV	0.93 (0.68, 1.00)	0.67 (0.22, 0.96)
NPV	0.83 (0.66, 0.93)	0.75 (0.53, 0.90)

**Table 3 diagnostics-13-01007-t003:** Skill scores for the radiological readers.

	Training Set (95% CI)	Test Set (95% CI)
Accuracy	0.42 (0.28, 0.57)	0.4 (0.23, 0.59)
Sensitivity	0.95 (0.75, 1.00)	0.80 (0.44, 0.97)
Specificity	0.07 (0.01, 0.22)	0.20 (0.06, 0.44)
Youden’s Index	0.02 (−0.24, 0.21)	0.00 (−0.50, 0.41)
PPV	0.40 (0.26, 0.56)	0.33 (0.16, 0.55)
NPV	0.67 (0.09, 0.99)	0.67 (0.22, 0.96)

## Data Availability

The datasets generated during and/or analyzed during the current study are available from the corresponding author upon reasonable request.

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
