# Peer review of "Freehand 1.5T MR-Guided Vacuum-Assisted Breast Biopsy (MR-VABB): Contribution of Radiomics to the Differentiation of Benign and Malignant Lesions"

_diagnostics, 2023, doi:10.3390/diagnostics13061007_

Round 1

Reviewer 1 Report

The research and analysis methods used in the article are consistent with the current similar research strategies, and there is no major problem.

The following points are worthy of the authors' attention:

1. It was mentioned in the discussion that there is currently no model for assessing benign and malignant breast images of patients and the necessity of VABB. Although there is no public and widely accepted model so far, similar studies have been carried out for some time , so this statement is problematic.

Please refer to https://www.ncbi.nlm.nih.gov/pmc/articles/PMC9026298/ and https://www.ncbi.nlm.nih.gov/pmc/articles/PMC8146084/

2. All the Radiomics features used in the initial analysis should be listed, as well as the specific features selected by statistical methods (such as regression function or yoden's index), as well as the corresponding p-value.

3. The AUC of this study is slightly behind the same type of research. The current AUC of the same type of research is about 0.7-0.72, and the number of participants is also small.

Author Response

  1. It was mentioned in the discussion that there is currently no model for assessing benign and malignant breast images of patients and the necessity of VABB. Although there is no public and widely accepted model so far, similar studies have been carried out for some time , so this statement is problematic.

Please refer to https://www.ncbi.nlm.nih.gov/pmc/articles/PMC9026298/ and https://www.ncbi.nlm.nih.gov/pmc/articles/PMC8146084/

We thank the referee for this comment. In the amended version of the paper, we have modified the original (surely too apodictic) sentence with a more critical comment pointing out that there is currently no public and widely accepted radiomics-based guideline for the pre-operative prediction of malignancy likelihood in patients amenable of MR-VABB. Further, we have added the two references suggested by the referee in the amended version of the manuscript's bibliography

  1. All the Radiomics features used in the initial analysis should be listed, as well as the specific features selected by statistical methods (such as regression function or yoden's index), as well as the corresponding p-value.

The reason why we did not include this information is because the number of features is really high. However, we agree with the referee that this is an important support to the comprehension of our analysis. Therefore, following https://www.ncbi.nlm.nih.gov/pmc/articles/PMC9026298/pdf/diagnostics-12-00771.pdf, Figure 2, we have added a heatmap of the correlation between pairs of radiomics features

  1. The AUC of this study is slightly behind the same type of research. The current AUC of the same type of research is about 0.7-0.72, and the number of participants is also small.

We thank the referee for this comment. In fact, our analysis obtains an AUC equal to 0.86 for the training set and to 0.73 for the test set (see Discussion). In the two papers mentioned in item 1, the AUC values are between 0.48 and 0.59 (first paper), and between 0.80 and 0.90 (second paper). However, in this latter study there is no separation between training and test sets, which typically leads to higher AUC values. We have added a comment on this in the amended version of the Discussion

Reviewer 2 Report

The authors tried to show that non-invasive radiomics analysis of DCE-MRI can avoid unnecessary MR-VABB in near future. This is an interesting article to the avid readers of Diagnostics. The authors should answer the following questions and revise the manuscript accordingly.

1) On what bases training set and validation set are divided?

2) How do you difference between gray intensity and contrast local intensity?

3) Acronyms (all) should be expanded at the first use in the text ex. PPV .

4) How do you take into account of blood vessel density in differentiating contrast obtained from DCE-MRI? 

5) Detailed experimental protocol of DCE-MRI should be mentioned in methods section.

Author Response

The authors tried to show that non-invasive radiomics analysis of DCE-MRI can avoid unnecessary MR-VABB in near future. This is an interesting article to the avid readers of Diagnostics. The authors should answer the following questions and revise the manuscript accordingly.

1) On what bases training set and validation set are divided?

The training set has been populated by maintaining the same balance as for the test set. Specifically, the training set includes 50 patients, with 20 patients with positive histology; the test set includes 30 patients, with 10 patients with positive histology. We did not consider any validation set, because the number of parameters to optimize in the regularization network is very small. We have added a comment on this in the methodology section

2) How do you difference between gray intensity and contrast local intensity?

The gray intensity denotes the absolute value of the pixel content, while the feature 'Contrast' is a Gray Level Co-occurrence Matrix (GLCM) features. Specifically, it is a weighted sum of the normalized co-occurence matrix entries.

3) Acronyms (all) should be expanded at the first use in the text ex. PPV .

Done

4) How do you take into account of blood vessel density in differentiating contrast obtained from DCE-MRI?

The highest peak of the enhanced phase in accordance with the time-intensity curve was selected on subtracted images to differentiate the lesion with high contrast reflecting neo-vascularization typical of breast lesions. The regions of interest were placed into the area of the lesion where the enhancement was strongest in the first non-subtracted postcontrast image.  When not possible, we selected the MRI acquisition with the highest peak of the enhanced phase (the first one of the second sequence when the highest peak was not reached at first). This method is consistent with recent literature [Li YZ, Huang YH, Su XY, et al. Breast MRI Segmentation and Ki-67 High- and Low-Expression Prediction Algorithm Based on Deep Learning. Comput Math Methods Med. 2022;2022:1770531. Published 2022 Oct 4. doi:10.1155/2022/1770531 and Attention-based Deep Learning for the Preoperative Differentiation of Axillary Lymph Node Metastasis in Breast Cancer on DCE-MRI]

5) Detailed experimental protocol of DCE-MRI should be mentioned in methods section.

We have not an experimental MRI protocol for this study since, for Radiomics analysis of MR-VABB lesions, data were collected retrospectively. The original study, from which we collected the retrospective data, was prospective and included both symptomatic and asymptomatic women, involving women with clinical findings or high-risk patients with BIRADS breast density C and D [NCT03033030] (see lines 97-99). DCE-MRI was done according to a standard clinical protocol including the sequences described in the paper (see lines 116-130)

Round 2

Reviewer 1 Report

Issues raised in previous reviews have been revised accordingly. I recommend this article for publication in its current form.